# Developing lifestyle intervention program for pre-hypertensive patients; consensus building using a modified Delphi approach

Danish Hassan[1]*, Syed Shakil Ur Rehman[1], Saira Khalid[1], Imran Tipu[2], Muhammad Husnain[1]

1 Riphah College of Rehabilitation & Allied Health Sciences, Riphah International University, Lahore, Pakistan, 2 School of Health Sciences, University of Management & Technology, Lahore, Pakistan

* Danish.hassan009@gmail.com

**Data Availability Statement:** All relevant data are within the manuscript and its Supporting Information files.

## Abstract

### Background

Prehypertension is a preclinical state of hypertension which leads to an increased likelihood of coronary heart disease, myocardial infarction, cerebrovascular disease as well as target organ damage. Addressing pre-hypertension through early lifestyle interventions is crucial to mitigating these detrimental effects and improving long-term health outcomes. So, the main objective of this study is to develop a lifestyle intervention program (LSIP) for the management of prehypertension using consensus building approach.

### Methods

It was a three round online modified Delphi study with 70 members panellists. All panellists had an experience of prehypertension either as patients (n = 30) or professionals (n = 40). Round 1 included initial recommendations developed from a previous systematic review and metanalysis, which were rated by panellists for their importance on a 5-point Likert scale. Panellists could also suggest additional items in the Round 1. Round 2 and 3 included all items from the Round 1 with new items suggested by the panellists. Data was analysed descriptively using SPSS version 29. All items receiving at least 70% of all respondents combined rating of 'Important' and 'Very Important' in Round 3 were included in the final set of recommendations.

### Results

Fifty-one panellists (80.9%) (patients = 25, professionals = 26) completed Round 3. Twenty-six recommendation items were included in the Round 1. Twenty new items were added in Round 2 with 46 total items in Round 2 and 3. Thirty-five of these items reached consensus in Round 3. The final set of recommendation comprised of 15 educational. 10 dietary, and 10 exercise recommendations.

**Funding:** This research was funded by Higher Education Commission, Pakistan under National Research Program for Universities (NRPU) (Ref No: 20-14875/NRPU/R&D/HEC/2021). HEC is a nonprofit government funded organization that supports research and development activities in Pakistan. The funders had no role in study design, data collection and analysis, decision to publish, or preparation of the manuscript.

**Competing interests:** The authors have declared that no competing interests exist.

## Conclusion

This modified Delphi study developed a comprehensive LSIP for the prevention of prehypertension, incorporating a holistic approach with educational, dietary, and exercise components aimed at the general population. Previously established standards of care (SOC) for managing prehypertension varied significantly and often provided fragmented guidance particularly on physical activity and education. This preventive model offers a novel and scalable approach for early intervention in prehypertension, potentially reducing reliance on medications and improving long-term health outcomes.

## Introduction

Joint National Committee 7[th] (JNC-7) report classified blood pressure in the range of 120-139/80-89 mm Hg as "Prehypertension" [1]. Prehypertension is now a days a common condition that affects population of wide age ranges, ethnicity, gender, and geography. The global prevalence of prehypertension ranges form 21–52% [2–15], burdening mostly developing and underdeveloped countries. Blood pressure within the prehypertension range is linked with higher incidence of hypertension and increased risk of cardiovascular disease, coronary heart disease, myocardial infarction [16] cerebrovascular disease [17], as well as target organ damage such as early atherosclerosis, microvascular damage, coronary artery calcification, vascular remodelling, and left ventricular hypertrophy [18]. Early detection and management of prehypertension is important to prevent hypertension-related complications and all-cause mortality.

Previous studies have narrated various modifiable risk factors for developing prehypertension that includes urban residential setting [19], single / alone living arrangements [20, 21], types of occupation, low wealth index [22], being active or past smoker [23], low physical activity, less sleep duration [24], high body mass index [25], high abdominal obesity [26], hyperglycaemia [22], high visceral adipose Index [25], and dyslipidaemia [23]. Diet has also a major contribution in occurrence of prehypertension with suboptimal percentage of carbohydrates, proteins and fats in diet [27], low fruits and vegetables consumption [22], high salt intake [28], alcohol consumption some notable risk factors. All the risk factors stated are attributed to the lifestyle which is living conditions behaviours or habits that are typical or chosen by a person or a community. Due the rapid pace of urbanization, automation, and modernization in the 20[th] in the past decade, the role lifestyle has been pivotal in manifestation of various non communicable disease [29]. JNC-7 has recommended lifestyle modification over pharmacotherapy in management of prehypertension and uncomplicated hypertension [30]. Lifestyle recommendations have also been previously reported for management of essential [31] and resistant hypertension [32, 33] but as an adjuvant to pharmacological intervention. However, no such intervention is available for secondary hypertension due to its pathophysiology requiring specifically targeted medical interventions. Pakistan ranks 5[th] in terms of world most populous country and 2[nd] in South Asian region [34]. Economic Survey of Pakistan (2020–21) states that the country only spends 1.2% of its GDP on healthcare, which is less than the WHO's recommended 5% [35]. The scarcity of available data on the prevalence of prehypertension makes it difficult to assess its burden. However, the high incidence of hypertension recorded in numerous surveys can provide an indication of its impact. National Health Survey of Pakistan (NHSP), WHO Stepwise approach to surveillance (STEPS) & National Diabetes Survey of

Pakistan (NDSP) estimates ≈19–46% Pakistani adults having hypertension, with higher prevalence in men and urban population [36, 37]. Significant proportion of patients presents with uncontrolled hypertension in in the medical emergency, accounting for more than one-fourth of all cases [38]. Several other factors also contribute to the difficulties in efficiently managing hypertension which include a lack of awareness regarding the seriousness of the disease, limited understanding, constraints within the healthcare system, inadequate health education, financial limitations, and religious beliefs [39].

Therefore, it is imperative to manage the preclinical stage of hypertension, known as prehypertension, to prevent additional strain on an already burdened healthcare system. So, the main objective of this study is to develop a lifestyle intervention program (LSIP) for the management of prehypertension using consensus building approach.

## Material & methods

This was three rounds Delphi study that was conducted online and in accordance with principles set forth in Accurate Consensus Reporting Document (ACCORD) checklist [40] (S1 Checklist) and proposed Delphi quality indicators [41]. The protocol for conductance of this study was approved by Research & Ethics Committee of Riphah International University, Lahore (Ref. No. REC/RCR & AHS/21/1101) campus prior to initiation of this study was not registered anywhere. Signed consent form was obtained from each participating member. The project advisory committee, comprising four members (a patient representative, a physical therapist, a nutritionist, and a cardiologist), oversaw the Delphi rounds. The professional members of the advisory committee had at least 10 years of experience in their relevant domain. Once consensus was reached, same four-member group also supervised the development of the LSIP.

A modified Delphi technique was used in this process of consensus building with the initial recommendation based on evidence from literature. The Modified Delphi method is a combination of the Nominal Group Technique (NGT) and the Delphi method. It optimizes the advantages of both consensus methods by first gathering information through questionnaires (consensus measurement) and then conducting a structured in-person meeting (consensus development) [42]. The approach has the benefit of reducing cognitive bias by lessening the burden on panellists [43].

### Delphi panellist

Delphi panellists consisted of professionals and representatives of the patient's population. Patients' population was screened from a free medical camp using purposive sampling technique. All included patients were adults diagnosed with prehypertension as per JNC 7 criteria [1] and able to communicate and comprehend in Urdu and English language and use/access the internet and email. Professional panellists included all the stake holders that were involved in the management of hypertension/hypertensive disorders as per the clinical paradigm practiced in Pakistan and were selected using snowball sampling from tertiary care hospitals across Lahore. All the expert panellists had at least 3 years of experience in their respective areas.

### Data collection

The data was collected from January 2022 to December 2022. Online survey tool–Google Form was used to collect the data from panellists and administered via email. Panellists were required to complete the consent statement prior to completing the survey. Reminders were provided via email and/or telephone to help maximize response rates. All individuals who completed Round 1 were subsequently emailed links to Rounds 2 and 3. The schematic process

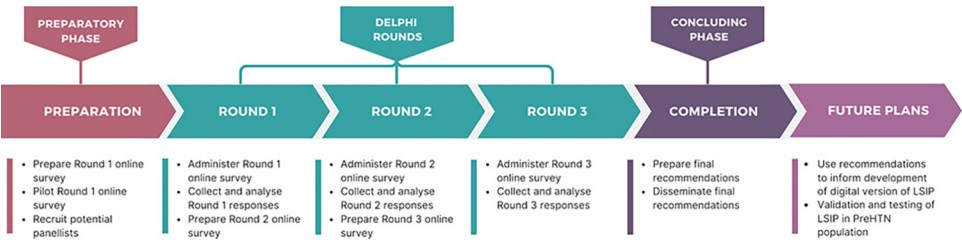

**Fig 1. Delphi process flow chart.**

of modified Delphi process is illustrated in Fig 1. Anonymity was assured for participants throughout the Delphi study. Responses were not shared among panellists, protecting their individual contributions.

**Round 1.** Round 1 included the initial set of recommendations in three areas; educational, dietary, and exercise recommendations form the extensive review of published literature. There recommendations were based on prior systematic review and metanalysis (PROSPERO registration number: CRD42023482512).

Round 1 also encompassed inquiries regarding the attributes of the panellists. Distinct question sets were provided for patient and professional panellists, concentrating on their socio-demographic and employment attributes, also addressing the clinical role and experience of the professional panellists. The panellists were instructed to assess the importance of each suggested item using a five-point Likert scale, ranging from 'Not at all important = 1' to 'Very important = 5'. For all items', detailed descriptions were provided to offer panellists comprehensive explanations. Each recommendation section concluded with the inclusion of open-ended question, which provided panellists with the opportunity to propose supplementary items.

Prior to conduction of Round 1, the survey was piloted on 05 experts (02 physical therapist, 01 nutritionist and 02 cardiologist) that led to minor wording changes for clarity in 08 recommendations items. None of the that participated in the pilot study joint the main panel of experts.

**Round 2.** In Round 2, all recommendation items from Round 1 were included to ensure each item had an equal chance of achieving a high level of consensus [44]. This method was chosen to prioritize the items based on the panellists' feedback in the final round. Each item from Round 1 was presented alongside three charts depicting the panellists' importance ratings from Round 1 (S1 Fig). Providing panellists with a summary of previous results is a well-established method for encouraging them to reconsider their initial judgments and foster consensus [44]. Additionally, round 2 incorporated new recommendation items generated from the open-ended responses in Round 1. Like Round 1, panellists were asked to rate the importance of each item using a five-point Likert scale. To reduce the burden on both panellists and researchers, no open-ended options were included in Round 2.

**Round 3.** Round 3 followed the same format as Round 2, including all the items from the previous round along with three charts summarizing the panellists' importance ratings from Round 2. As in the earlier rounds, panellists were asked to evaluate the significance of each item using a five-point Likert scale. Consensus was determined by items receiving at least 70% combined ratings of "important" and "very important" from the panellists.

## Development of LSIP

After completing the three rounds of the Delphi process, the items that reached consensus were reviewed by the project advisory committee. During this review, any duplicate items

were removed to streamline the recommendations. We adhered to the guidelines outlined by Hoffman et al., [45] for developing impactful health educational materials. This included aiming for a sixth to seventh grade reading level, employing short sentences conveying single ideas, utilizing common language, avoiding jargon or abbreviations, adopting an active voice and conversational tone, and addressing the reader directly in the second person. For the nutritional component, alternatives to consensus items were selected based on seasonal and geographical availability to ensure accessibility for most of the population across the globe. Similarly, the exercise recommendations focused on alternate activities that were easy to perform and did not require any special equipment, making them suitable for a wide range of individuals. This careful selection process ensured that the LSIP was both feasible and effective for participants.

## Data analysis

Data was analysed descriptively using Microsoft 365 Excel and IBM SPSS Statistics version 29. While there are established guidelines on conducting Delphi studies [41], there is no universally accepted threshold for defining consensus; however, a commonly used threshold is 70% [46–48]. Hence, a cut-off value of 70% level of agreement was chosen a priori as the consensus threshold [49]. Consensus was reached if any item received at least 70% combined ratings of 'important' and 'very important' from the panellists. All items that reached consensus in Round 3, considering all respondents together, were included in the final set of recommendations. For readability assessment, the Flesch Reading Ease test and Flesch-Kincaid Grade Level test were used [50].

## Results

The developed LSIP was rated as standard in reading difficulty with Flesch Reading Ease test score of score of 65. Flesch-Kincaid Grade Level for the LSIP was 7.29 targeting a person with educational equivalent to 7th grade (age range 12–13 years). Time required for panellists to complete the consensus surveys in the Delphi study varied across the three rounds. In Round 1, each panellist took approximately 9 to 13 minutes to complete the survey while Round 2 & 3 took about 15 to 23 minutes per panellist due to the increased number of items and the additional task of reviewing summary charts from the previous round.

### Delphi panellist

Ninety-three panellists were screened for Round 1 and emailed the link for the first round of the Delphi study. Among the initially screened panellists, only 70 (75.2%) responded, including 30 patients and 40 professionals. In Round 2 of the Delphi study, 63 (90%) panellists responded, which were further reduced to 51 (80.9%) in Round 3 (Table 1). The decrease in responses was solely due to the non-response rate from the panellists. Demographic details of the patients and professionals are provided in Tables 2 and 3.

### Importance ratings overview

Based on the initial literature review, 26 recommendation items were included in Round 1. Consensus was reached for 18 items among professionals and 12 items among patients, with a total of 18 items achieving consensus in the first round. Twenty new items were added by panellists in Round 2, resulting in a total of 46 items. In Round 2, 32 items reached consensus overall, with 25 items receiving consensus from patients and 38 items from professionals. By Round 3, three additional items, including those from Round 2, reached

**Table 1. Response rate and composition of panellists by Delphi rounds.**

| Panellist group | Characteristics of panellists | Round 1 | Round 2 | Round 3 |
|---|---|---|---|---|
| | | N = 93 | N = 70 | N = 63 |
| | | Responded | Responded | Responded |
| | | N (%) | N (%) | N (%) |
| Patient | Patient diagnosed with Pre-Hypertension | 30 (32.2) | 28 (40.0) | 25 (39.6) |
| Professional | Physical Therapist | 10 (10.8) | 9 (12.9) | 7 (11.1) |
| | Nutritionist / Dietitian | 9 (9.7) | 7 (10.0) | 6 (9.5) |
| | General Physician | 8 (8.6) | 7 (10.0) | 5 (7.9) |
| | Cardiologist | 8 (8.6) | 8 (11.4) | 5 (7.9) |
| | Nutritionist / Dietitian with Certified Lifestyle Practitioner | 5 (5.4) | 4 (5.7) | 3 (4.8) |
| **Did not respond** | | **23 (24.7)** | **7 (10.0)** | **12 (19.0)** |
| **Total response rate** | | **70 (75.2)** | **63 (90.0)** | **51 (80.9)** |

consensus, bringing the overall total to 35 items. Specifically, in Round 3, 24 items reached consensus among patients, while professionals reached consensus on 37 items in total (Table 4).

**Table 2. Patient panellist characteristics.**

| | Patient Panellist Characteristics (N = 30) n (%) |
|---|---|
| **Living Location** | |
| Lahore | 16 (53.3) |
| Gujranwala | 4 (13.3) |
| Sialkot | 5 (16.6) |
| Peshawar | 1 (3.3) |
| Islamabad | 4 (13.3) |
| **Gender** | |
| Male | 11 (36.6) |
| Female | 19 (63.3) |
| **Age Category** | |
| Less than 25 years | 2 (6.6) |
| Between 26–40 years | 24 (80.0) |
| Between 41–60 years | 4 (13.3) |
| **Highest level of qualification** | |
| Middle | 1 (3.3) |
| Matriculation (SSC) | 1(3.3) |
| Intermediate (HSSC) | 1 (3.3) |
| Vocational | 5 (16.6) |
| Graduate Degree | 9 (30) |
| Postgraduate Degree | 12 (40) |
| Doctorate | 1 (3.3) |
| **Current employment status** | |
| Unemployed | 3 (10) |
| Self employed | 1 (3.3) |
| Employed (Full or Part Time) | 21 (70) |
| Home Maker | 5 (16.6) |

[SSC: Secondary School Certificate, HSSC: Higher Secondary School Certificate]

**Table 3. Professional panellist characteristics.**

|  | Professional Panellist Characteristics (N = 40) n (%) |
|---|---|
| **Working Location** | |
| Lahore | 40 (100) |
| **Highest level of qualification** | |
| FCPS / MD | 16 (40.0) |
| Masters | 21 (52.5) |
| PhD | 3 (7.5) |
| **What setting(s) do you currently work in? (Select all that apply)** | |
| Government hospital | 6 (15.0) |
| Private hospital | 15 (37.5) |
| University | 11 (27.5) |
| Self-owned clinic / Medical center | 8 (20.0) |
| **Do you currently provide clinical care to patients diagnosed with PreHTN?** | |
| Yes | 10 (25.0) |
| No | 30 (75.0) |

[FCPS: Fellow of College of Physicians and Surgeons, MD: Doctor of Medicine, PhD: Doctor of Philosophy]

## Educational recommendations overview

Based on the initial literature review, 12 educational recommendation items were included in Round 1. Consensus was reached for 9 items among professionals and 7 items among patients, with a total of 8 items achieving consensus in this round (S1 Table). In Round 2, 19 educational recommendation items were evaluated, resulting in 14 items reaching consensus overall, with 12 items receiving consensus from patients and 16 from professionals. By Round 3, 15 educational items reached consensus, with 12 items receiving consensus from patients and 14 from professionals (Table 5).

## Dietary recommendations overview

Eight dietary recommendation items were initially included in Round 1. Consensus was reached for 6 items among professionals and 2 items among patients, with a total of 6 items

**Table 4. Number of recommendation items.**

| Number of recommendation items | | | Educational Recommendations | Dietary Recommendations | Exercise Recommendations | Total Recommended Items |
|---|---|---|---|---|---|---|
| Round 1 | Total Items | | 12 | 8 | 6 | 26 |
|  | Reached consensus | Patients | 7 | 2 | 3 | 12 |
|  |  | Professionals | 9 | 6 | 3 | 18 |
|  |  | All Panellists | 8 | 6 | 4 | 18 |
| Round 2 | Total Items | | 19 | 14 | 13 | 46 |
|  | Reached consensus | Patients | 12 | 7 | 6 | 25 |
|  |  | Professionals | 16 | 13 | 9 | 38 |
|  |  | All Panellists | 14 | 10 | 8 | 32 |
| Round 3 | Total Items | | 19 | 14 | 13 | 46 |
|  | Reached consensus | Patients | 12 | 6 | 6 | 24 |
|  |  | Professionals | 14 | 12 | 11 | 37 |
|  |  | All Panellists | 15 | 10 | 10 | 35 |

**Table 5. Educational recommendations: Importance ratings summary.**

| Total Educational Recommendations items | Round 1 (n = 70) | | | Round 2 (n = 63) | | | Round 3 (n = 51) | | |
|---|---|---|---|---|---|---|---|---|---|
| | % Very Important rating | % Important rating | % Consensus | % Very Important rating | % Important rating | % Consensus | % Very Important rating | % Important rating | % Consensus |
| **1. Smoking Cessation** | 50.0 | 41.4 | 91.4 | 52.4 | 38.1 | 90.5 | 49.0 | 39.2 | 88.2 |
| **2. Physical Activity** | 48.6 | 45.7 | 94.3 | 50.8 | 46.0 | 96.8 | 51.0 | 45.1 | 96.1 |
| **3. Stress Management** | 71.4 | 28.6 | 100.0 | 69.8 | 30.2 | 100.0 | 68.6 | 31.4 | 100.0 |
| **4. Sleep Duration** | 57.1 | 28.6 | 85.7 | 57.1 | 34.9 | 92.1 | 56.9 | 35.3 | 92.2 |
| **5. Salt Intake** | 100.0 | 0.0 | 100.0 | 100.0 | 0.0 | 100.0 | 100.0 | 0.0 | 100.0 |
| 6. Soft Drink Consumptions | 30.0 | 30.0 | 60.0 | 28.6 | 27.0 | 55.6 | 27.5 | 29.4 | 56.9 |
| 7. Alcohol Consumption | 27.1 | 31.4 | 58.6 | 30.2 | 30.2 | 60.3 | 29.4 | 25.5 | 54.9 |
| **8. Obesity** | 51.4 | 38.6 | 90.0 | 49.2 | 39.7 | 88.9 | 49.0 | 39.2 | 88.2 |
| **9. Blood Glucose Level** | 30.0 | 31.4 | 61.4 | 38.1 | 31.7 | 69.8 | 41.2 | 31.4 | 72.5 |
| 10. Blood Uric Acid Level | 11.4 | 25.7 | 37.1 | 11.1 | 23.8 | 34.9 | 13.7 | 23.5 | 37.3 |
| **11. Blood Lipid Level** | 30.0 | 41.4 | 71.4 | 36.5 | 47.6 | 84.1 | 35.3 | 45.1 | 80.4 |
| **12. Blood High Density Lipoproteins Level** | 55.7 | 38.6 | 94.3 | 55.6 | 39.7 | 95.2 | 52.9 | 41.2 | 94.1 |
| **13. General Definition–Prehypertension** | N/A | | | 28.6 | 42.9 | 71.4 | 31.4 | 45.1 | 76.5 |
| **14. Diagnosis of Prehypertension** | N/A | | | 57.1 | 23.8 | 81.0 | 56.9 | 27.5 | 84.3 |
| **15. Burden of Prehypertension** | N/A | | | 71.4 | 11.1 | 82.5 | 66.7 | 11.8 | 78.4 |
| **16. After effects of Prehypertension** | N/A | | | 52.4 | 28.6 | 81.0 | 52.9 | 27.5 | 80.4 |
| 17. Ethnicity | N/A | | | 15.9 | 28.6 | 44.4 | 15.7 | 25.5 | 41.2 |
| **18. Aging** | N/A | | | 60.3 | 22.2 | 82.5 | 62.7 | 19.6 | 82.4 |
| **19. Family History** | N/A | | | 25.4 | 55.6 | 81.0 | 27.5 | 52.9 | 80.4 |

[Items in bold reached consensus at the end of Round 3]

achieving consensus in this round (S2 Table). In Round 2, 14 dietary recommendation items were evaluated, resulting in 10 items reaching consensus overall, with 7 items receiving consensus from patients and 13 from professionals. By Round 3, 10 dietary items reached consensus, with 6 items receiving consensus from patients and 12 from professionals (Table 6).

## Exercise recommendations overview

Six exercise recommendation items were included in Round 1. Consensus was reached for 3 items among both professionals and patients, with a total of 4 items achieving consensus in this round (S3 Table). In Round 2, 13 exercise recommendation items were evaluated, resulting in 8 items reaching consensus overall, with 6 items receiving consensus from patients and 9 from professionals. By Round 3, 10 exercise items reached consensus, with 6 items receiving consensus from patients and 11 from professionals (Table 7).

## Development of LSIP

Through consensus on dietary and exercise recommendations, the project advisory committee facilitated the development of the LSIP. Redundant recommendation items were eliminated, and alternative food items and exercises were suggested to enhance the program's wider applicability and generalizability.

Table 6. Dietary recommendations: Importance ratings summary.

| Total Dietary Recommendations items | Round 1 (n = 70) | | | Round 2 (n = 63) | | | Round 3 (n = 51) | | |
|---|---|---|---|---|---|---|---|---|---|
| | % Very Important rating | % Important rating | % Consensus | % Very Important rating | % Important rating | % Consensus | % Very Important rating | % Important rating | % Consensus |
| **1. Olive Oil** | 41.4 | 34.3 | 75.7 | 44.4 | 36.5 | 81.0 | 43.1 | 35.3 | 78.4 |
| **2. Green Tea** | 37.1 | 38.6 | 75.7 | 41.3 | 38.1 | 79.4 | 39.2 | 37.3 | 76.5 |
| **3. Omega-3 fat alpha-linolenic acid (ALA)** | 64.3 | 21.4 | 85.7 | 61.9 | 23.8 | 85.7 | 60.8 | 21.6 | 82.4 |
| 4. High Intake of Dietary Potassium | 34.3 | 27.1 | 61.4 | 38.1 | 25.4 | 63.5 | 35.3 | 19.6 | 54.9 |
| 5. Nitric Oxide | 30.0 | 34.3 | 64.3 | 30.2 | 39.7 | 69.8 | 33.3 | 33.3 | 66.7 |
| **6. Fruits Consumption** | 40.0 | 31.4 | 71.4 | 49.2 | 34.9 | 84.1 | 47.1 | 31.4 | 78.4 |
| **7. Vegetables Consumptions** | 37.1 | 40.0 | 77.1 | 44.4 | 39.7 | 84.1 | 41.2 | 39.2 | 80.4 |
| **8. Nuts Consumptions** | 41.4 | 31.4 | 72.9 | 46.0 | 31.7 | 77.8 | 37.3 | 37.3 | 74.5 |
| **9. Low Fat Dairy Products** | N/A | | | 39.7 | 42.9 | 82.5 | 41.2 | 43.1 | 84.3 |
| **10. Whole Grains** | N/A | | | 34.9 | 38.1 | 73.0 | 33.3 | 37.3 | 70.6 |
| 11. Oats | N/A | | | 25.4 | 44.4 | 69.8 | 25.5 | 41.2 | 66.7 |
| **12. Barley** | N/A | | | 20.6 | 50.8 | 71.4 | 19.6 | 52.9 | 72.5 |
| **13. Cereals** | N/A | | | 41.3 | 46.0 | 87.3 | 43.1 | 43.1 | 86.3 |
| 14. Pre-Breakfast | N/A | | | 15.9 | 20.6 | 36.5 | 15.7 | 23.5 | 39.2 |

[Items in bold reached consensus at the end of Round 3]

For instance, isometric hand grip exercises and resistance exercises were grouped as one category. Aerobic exercises, including brisk walking, desk treadmilling, were regrouped under the category of aerobic exercises. Stretching and yoga exercises were recommended as warm-

Table 7. Exercise recommendations: Importance ratings summary.

| Total Exercise Recommendations items | Round 1 (n = 70) | | | Round 2 (n = 63) | | | Round 3 (n = 51) | | |
|---|---|---|---|---|---|---|---|---|---|
| | % Very Important rating | % Important rating | % Consensus | % Very Important rating | % Important rating | % Consensus | % Very Important rating | % Important rating | % Consensus |
| **1. Yoga Therapy** | 30.0 | 40.0 | 70.0 | 25.4 | 57.1 | 82.5 | 21.6 | 60.8 | 82.4 |
| **2. Isometric Hand Grip Exercises** | 35.7 | 27.1 | 62.9 | 33.3 | 38.1 | 71.4 | 37.3 | 39.2 | 76.5 |
| **3. Aerobic Exercise** | 28.6 | 47.1 | 75.7 | 25.4 | 49.2 | 74.6 | 23.5 | 54.9 | 78.4 |
| **4. Stretching Exercises** | 35.7 | 31.4 | 67.1 | 36.5 | 33.3 | 69.8 | 35.3 | 35.3 | 70.6 |
| **5. Resistance Exercises** | 34.3 | 44.3 | 78.6 | 34.9 | 44.4 | 79.4 | 35.3 | 43.1 | 78.4 |
| **6. Walking** | 31.4 | 41.4 | 72.9 | 30.2 | 41.3 | 71.4 | 31.4 | 41.2 | 72.5 |
| 7. Commuting | N/A | | | 20.6 | 44.4 | 65.1 | 19.6 | 45.1 | 64.7 |
| **8. Brisk Walking** | N/A | | | 28.6 | 49.2 | 77.8 | 27.5 | 52.9 | 80.4 |
| **9. Desk treadmilling** | N/A | | | 28.6 | 41.3 | 69.8 | 35.3 | 41.2 | 76.5 |
| 10. Swimming | N/A | | | 19.0 | 25.4 | 44.4 | 19.6 | 23.5 | 43.1 |
| 11. Hiking | N/A | | | 23.8 | 36.5 | 60.3 | 25.5 | 37.3 | 62.7 |
| **12. High Intensity Interval Training** | N/A | | | 46.0 | 25.4 | 71.4 | 56.9 | 25.5 | 82.4 |
| **13. Circuit Training** | N/A | | | 39.7 | 31.7 | 71.4 | 52.9 | 31.4 | 84.3 |

[Items in bold reached consensus at the end of Round 3]

up and cool-down routines. Both aerobic and resistance exercises were based upon the principles of interval training, incorporating alternate rest and exercise periods. Exercises using equipment such as cycle ergometers and resistance exercises with free weights were substituted with calisthenic or bodyweight exercises that target the same heart rate reserve and muscle groups. This approach ensures that LSIP remains both practical and evidence based.

For dietary recommendations, the project advisory committee streamlined the suggestions by removing redundant items with similar nutritional content. They adopted the concept of a healthy plate, creating a sample daily meal plan with specific portion sizes for each food group. Additionally, commercially available supplements, like omega-3 capsules, were replaced with recommendations for incorporating whole foods rich in omega-3 fatty acids, such as fatty fish and flaxseeds. This approach promotes a more practical and sustainable dietary approach.

## Discussion

The main objective of the study was to develop a consensus based LSIP for the management of prehypertension. The LSIP developed in the current research for prehypertension differs from previous SOC approaches [51–56] by targeting multiple aspects of lifestyle (educational, dietary, and exercise) rather than focusing on just one. Of the 35 recommendation items that reached consensus in Round 3 educational / lifestyle recommendations shared the largest proportion followed by equal proportion of dietary and exercise recommendations. The level of agreement between the patient and professional members of the panellists was greatest for educational recommendations topic section followed by exercise and dietary recommendation topics respectively.

Bulk of the educational/lifestyle recommendations from the panellists signify its role in prevention of prehypertension. Previous researches have also signified the importance of education in reductions of blood pressure. [57, 58]. High percentage of consensus ranging from 90 to 100% was obtained for physical activity, stress management, sleep duration, salt intake, and high-density lipoprotein level. The high prevalence of hypertension in low- to middle-income countries [59] necessitates delivering educational guidelines to patients and these recommendations are among the few commonly provided for both the prevention and management of hypertension. Overall, the level of consensus achieved was similar for both patient and professional panellists' members except for low consensus percentage for blood glucose level, common blood lipid level, general definition of prehypertension for patients. Although previous studies [21, 60] had listed recommendation as strong risk factor for developing prehypertension, health literacy is pivotal in management of NCDs [61]. A previous study also reported inadequate heath literacy regarding NCDs in South Asian region emphasizing the need to design and implement interventions focusing on health education of general population [62]. No consensus was reached for soft drink consumption, alcohol consumption, blood uric acid level, and ethnicity by both panellists' members of this Delphi study. In Pakistan, cultural and religious beliefs strongly influence views on no alcohol consumption irrespective of prehypertension management, making it non-significant for the consensus [63]. These factors highlight the need for targeted educational efforts to bridge the knowledge gap and achieve a more uniform consensus across different population groups in Pakistan. Based on consensus, the educational recommendations in the LSIP were broadly categorized into two sections: one covered general education on hypertension, including its burden, diagnosis, and aftereffects of prehypertension. The second section outlined risk factors identified as exclusively specific to prehypertension, which had previously only been established for hypertension.

Exercise has been advocated as one of best nonpharmacological interventions in mitigating high blood pressure in the past, however the type of exercise remains a matter of debate [64].

Though consensus was reached among patients for commuting but was excluded from the final recommendation due to low level of agreement among professional and vice vera for hiking. No consensus was reached for swimming by both patients and panellist as previous studies also reported indirect effects swimming on blood pressure reduction [65, 66]. Among the item that reached consensus were broadly categorised under the umbrella of relaxation, stretching, aerobic and strengthening exercises. Previous studies have emphases the role of relaxation exercises in blood pressure reduction [67–69] with yoga incorporating different relaxation techniques [70, 71] and stretching an effective intervention to reduce arterial stiffness, HR, and DBP [72]. Aerobic exercises have widely been recommended to prevent/treat hypertension [73, 74] ranging in intensity form low to high or either alternating high with low intensity or in the form of circuit that incorporates daily performed activities. Isometric resistance training is also advocated as one of the effective methods of managing increased blood pressure [75, 76]. Although the combined effects of aerobic and resistance exercise are deemed more beneficial in previous studies [77, 78], nevertheless resistance exercise remain effective alone also [79]. Based on consensus, the exercise recommendations in the LSIP were an amalgam of aerobic, strengthening, and stretching exercises. All exercises were calisthenic, time-efficient, required no equipment, could be done at home, and could be started immediately, making them accessible and beneficial for a wide range of the population.

In dietary recommendation no agreement was achieved for high intake of dietary potassium, nitric oxide, oats, and pre breakfast in round 3. Disparity existed between patients and professionals for recommendation of nitric oxide and oats, professionals being agreed but patients was not. The included dietary recommendations emphasize increased consumption of vegetables, fruits, whole grains, low-fat dairy products, and nuts. Additionally, olive oil consumption is included as a recommended component. The role of OMEGA 3 fats in cardiovascular health and prevention of cardiovascular diseases is well established [80] and was also included in the final recommendations. Omega-3 fatty acids, including ALA, EPA, and DHA, are known to reduce inflammation, lower triglyceride levels, and decrease the risk of heart disease [80–83]. In Pakistan, the daily diet predominantly features omega-6 oils, commonly found in affordable options like sunflower and soybean oils. However, the typical diet often lacks sufficient omega-3 fatty acids, which are crucial for overall cardiovascular health. A low intake of omega-3 fatty acids compared with omega-6s may contribute to inflammation and chronic diseases, such as rheumatoid arthritis, diabetes, atherosclerosis, and heart failure [84, 85]. Omega-3s, particularly alpha-linolenic acid (ALA), are abundantly present in seeds such as flaxseeds and chia seeds, yet these are rarely consumed. EPA and DHA, vital omega-3 fatty acids, are primarily found in marine fish that are not commonly consumed in Pakistan. Therefore, the focus should be on increasing the intake of OMEGA 3 fats, incorporating more seeds and other fish into the diet can help improve cardiovascular health and reduce the risk of hypertension and other cardiovascular diseases. Following consensus, the dietary recommendations in the LSIP were based on the DASH diet and incorporated some elements from the Mediterranean diet, utilizing the healthy plate concept. To enhance patient convenience, the LSIP included a categorized list of food items, ranging from most recommended to least recommended.

Although the Delphi process facilitated a structured and iterative approach to developing the LSIP, ensuring consensus among experts, there were few limitations of the study. The study population consisted entirely of Asian panellists and patients. This may limit the generalizability of the recommendations due to potential racial disparities in dietary needs and preferences. Open-ended questions were only included in the first round of the Delphi study. This may have limited the opportunity for panellists to provide nuanced feedback and explore alternative approaches in subsequent rounds. Moreover, the patient population necessarily

required technological literacy and bilingualism in English and Urdu as the consensus was done digitally and in English since mother language in Pakistan is Urdu. This could potentially exclude patients with lower literacy levels or those who are not comfortable with English, leading to a sample that may not be fully representative of the broader patient population in Pakistan.

## Conclusion

This modified Delphi study developed a comprehensive set of recommendations for managing prehypertension that inculcated education, dietary and exercise recommendation for adopting the healthier lifestyle. The LSIP is useful resource for health professionals in preventing prehypertension. This approach could be particularly beneficial in low- and middle-income countries, providing a scalable model for early intervention in prehypertension management while reducing the reliance on pharmacological interventions.

The LSIP, with its focus on education, diet, and exercise, provides a feasible strategy to address prehypertension, potentially reducing the risk of progression to full-blown hypertension and associated cardiovascular diseases. In the next phase of the this study the LSIP will be tested for its effects on cardiovascular, physical, biochemical and respiratory parameters in prehypertension patients.

## Supporting information

**S1 Checklist. ACCORD checklist.**
(DOCX)

**S1 Fig. Template for panellists' importance ratings.**
(TIF)

**S1 Table. Consensus percentage for patients and professional for educational recommendation items.**
(DOCX)

**S2 Table. Consensus percentage for patients and professional for dietary recommendation items.**
(DOCX)

**S3 Table. Consensus percentage for patients and professional for exercise recommendation items.**
(DOCX)

**S1 Dataset.**
(XLSX)

## Author Contributions

**Conceptualization:** Danish Hassan, Syed Shakil Ur Rehman.

**Data curation:** Imran Tipu, Muhammad Husnain.

**Formal analysis:** Danish Hassan.

**Funding acquisition:** Danish Hassan, Syed Shakil Ur Rehman.

**Investigation:** Imran Tipu.

**Methodology:** Danish Hassan, Imran Tipu.

**Project administration:** Syed Shakil Ur Rehman, Saira Khalid.

**Resources:** Saira Khalid.

**Software:** Saira Khalid.

**Validation:** Danish Hassan.

**Writing – original draft:** Danish Hassan, Muhammad Husnain.

**Writing – review & editing:** Danish Hassan, Muhammad Husnain.

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
