## [Editor Report · Decision Letter 0]

21 Jun 2024

PONE-D-24-23230Developing lifestyle intervention program for pre-hypertensive patients; consensus building using a modified Delphi approachPLOS ONE

Dear Dr. Hassan,

Thank you for submitting your manuscript to PLOS ONE. After careful consideration, we feel that it has merit but does not fully meet PLOS ONE’s publication criteria as it currently stands. Therefore, we invite you to submit a revised version of the manuscript that addresses the points raised during the review process.

Dear Authors,Instead of sending the manuscript for peer review, I am recommending few minor changes, which I feel could improve the messages conveyed through your work. Please revise accordingly so it may get a proper feedback from the reviewers.==============================

We look forward to receiving your revised manuscript.

Kind regards,

Yogesh Kumar Jain, MPH

Academic Editor

PLOS ONE

Journal Requirements:

Additional Editor Comments:

Dear Authors,

Upon careful reading, I could see that some substantial work has been done in development of a model with modifications in existing SOC for hypertensive patients. However, I would like to highlight that in the current structure, any potential reviewer might feel that the study is not novel and provides inferences that are already known to the research community. I recommend that you submit a modified version considering the following points to avoid rejection in the peer review process:

1. Modify the abstract - shorten the background, and in results/conclusion section include what novel implications the study adds. You may want to include the what educational, dietary, and exercise components came out of the study that were different from the earlier standard of care.

2. In the discussion section, please clearly mention what the final model includes, what is different from the earlier known standard - you might include some references for the previously know and used Standard of Care.

---

## [Author Response · Author response to Decision Letter 0]

1 Jul 2024

Comment: Please ensure that your manuscript meets PLOS ONE's style requirements, including those for file naming. 

Response: The manuscript has been formatted to meet PLOS ONE's style requirements, including those for file naming.

Comment: We note that the grant information you provided in the ‘Funding Information’ and ‘Financial Disclosure’ sections do not match. 

Response: The grant information provided in the ‘Funding Information’ and ‘Financial Disclosure’ sections now match.

Comment: Your ethics statement should only appear in the Methods section of your manuscript. If your ethics statement is written in any section besides the Methods, please move it to the Methods section and delete it from any other section. Please ensure that your ethics statement is included in your manuscript, as the ethics statement entered into the online submission form will not be published alongside your manuscript. Response: Ethics statement is not added in the methodology section and removed from other parts of the manuscript. Page: 5 Line no. 100

Comment: Please include captions for your Supporting Information files at the end of your manuscript, and update any in-text citations to match accordingly.

Response: The captions for the Supporting Information files have been included at the end of the manuscript, and all in-text citations have been updated accordingly. Page: 28

Comment: Modify the abstract - shorten the background, and in results/conclusion section include what novel implications the study adds. You may want to include the what educational, dietary, and exercise components came out of the study that were different from the earlier standard of care. Response: Abstract has been updated as per recommendations. Background is shorter now and results and conclusion sections are also updated. 

Page: 2, 3 Line no. 15 – 19, 38 – 50.

Comment: In the discussion section, please clearly mention what the final model includes, what is different from the earlier known standard - you might include some references for the previously know and used Standard of Care. 

Response: The discussion section has been modified, and additional text has been added to clarify the novelty of the final model compared to previous literature. References to the previously known and used Standard of Care have also been included. Page: 16 – 19

---

## [Decision Letter · Decision Letter 1]

17 Sep 2024

PONE-D-24-23230R1Developing lifestyle intervention program for pre-hypertensive patients; consensus building using a modified Delphi approachPLOS ONE

Dear Dr. Hassan,

Thank you for submitting your manuscript to PLOS ONE. After careful consideration, we feel that it has merit but does not fully meet PLOS ONE’s publication criteria as it currently stands. Therefore, we invite you to submit a revised version of the manuscript that addresses the points raised during the review process.

Dear Authors, thank you for the patience. The delay in decision was due to difficulty in securing reviewers for the manuscript. Nevertheless, during this round of review, we have received an overall positive feedback with very minor suggestion from one reviewer.

You are requested to submit the revision at the earliest with point wise reply and clarification, so we may make the decision directly at the editorial level.

We look forward to receiving your revised manuscript.

Kind regards,

Yogesh Kumar Jain, MPH

Academic Editor

PLOS ONE

Journal Requirements:

Additional Editor Comments:

Provided above.

Reviewers' comments:

Reviewer's Responses to Questions

**Comments to the Author**

1. If the authors have adequately addressed your comments raised in a previous round of review and you feel that this manuscript is now acceptable for publication, you may indicate that here to bypass the “Comments to the Author” section, enter your conflict of interest statement in the “Confidential to Editor” section, and submit your "Accept" recommendation.

Reviewer #1: (No Response)

Reviewer #2: All comments have been addressed

2. Is the manuscript technically sound, and do the data support the conclusions?

Reviewer #1: Yes

Reviewer #2: Yes

3. Has the statistical analysis been performed appropriately and rigorously? 

Reviewer #1: Yes

Reviewer #2: Yes

4. Have the authors made all data underlying the findings in their manuscript fully available?

Reviewer #1: Yes

Reviewer #2: Yes

5. Is the manuscript presented in an intelligible fashion and written in standard English?

Reviewer #1: Yes

Reviewer #2: Yes

6. Review Comments to the Author

Reviewer #1: (No Response)

Reviewer #2: Thank you for submitting your valuable work i would like to recommend adding some detailes on secondary hypertention on your introduction or discussion for example patients with renal hypertention and weather it had been addressed befor in other studies or not and if the same recommendations is applicable to patients with essential hypertention.

7. PLOS authors have the option to publish the peer review history of their article (what does this mean?). If published, this will include your full peer review and any attached files.

Reviewer #1: No

Reviewer #2: **Yes: **Seham elazab

---

## [Author Response · Author response to Decision Letter 1]

19 Sep 2024

1. Journal Requirements:

Author Response: The reference list is complete and is not being updated in the revised manuscript. No retracted paper has been cited in this manuscript to date.

2. Thank you for submitting your valuable work i would like to recommend adding some details on secondary hypertension on your introduction or discussion for example patients with renal hypertension and weather it had been addressed before in other studies or not and if the same recommendations is applicable to patients with essential hypertension. 

Author Response: Thank you for your valuable feedback and thoughtful recommendation. We appreciate your suggestion to include information on secondary hypertension, particularly in relation to conditions such as renal hypertension, and whether similar recommendations have been addressed in previous studies.

As the primary focus of our study is prehypertension, we would like to clarify that the lifestyle modifications recommended through our Delphi study are specifically designed for pre-hypertensive patients. According to JNC 7 guidelines for managing high blood pressure, prehypertensive patients are ideal candidates for lifestyle modification, potentially bypassing the need for pharmacological intervention. In contrast, the pathophysiology of essential and secondary hypertension is more complex and often associated with multiple comorbid factors. For example, renal hypertension may require specific medical interventions to address underlying kidney issues, whereas essential hypertension often necessitates pharmacological treatment.

We hope this explanation addresses your concern.

---

## [Editor Report · Decision Letter 2]

22 Sep 2024

PONE-D-24-23230R2Developing lifestyle intervention program for pre-hypertensive patients; consensus building using a modified Delphi approachPLOS ONE

Dear Dr. Hassan,

Thank you for submitting your manuscript to PLOS ONE. After careful consideration, we feel that it has merit but does not fully meet PLOS ONE’s publication criteria as it currently stands. Therefore, we invite you to submit a revised version of the manuscript that addresses the points raised during the review process.

Please see comments to authors below.

We look forward to receiving your revised manuscript.

Kind regards,

Yogesh Kumar Jain, MPH

Academic Editor

PLOS ONE

Journal Requirements:

Additional Editor Comments:

Dear Author, it seems that you have not made any changes in the re-submitted manuscript!

The uploaded version with tracked changes do not show any such changes being made. If this file has been uploaded by mistake, then you are requested to kindly upload the correct version at the earliest, and I would emphasise that you take caution, as is takes a lot of time for the editorial team as well as the reviewers to put in their comments. Such errors might prompt the reviewer to post a "Reject" recommendation, which would then be difficult to reverse from our end.

---

## [Author Response · Author response to Decision Letter 2]

23 Sep 2024

Response: The reference list is complete and is not being updated in the revised manuscript. No retracted paper has been cited in this manuscript to date.

2. Thank you for submitting your valuable work i would like to recommend adding some details on secondary hypertension on your introduction or discussion for example patients with renal hypertension and weather it had been addressed before in other studies or not and if the same recommendations is applicable to patients with essential hypertension.

Response: Thank you for your valuable feedback and thoughtful recommendation. We appreciate your suggestion to include information on secondary hypertension, particularly in relation to conditions such as renal hypertension, and whether similar recommendations have been addressed in previous studies.

As the primary focus of our study is prehypertension, we would like to clarify that the lifestyle modifications recommended through our Delphi study are specifically designed for pre-hypertensive patients. These interventions, which include exercise, diet, and other lifestyle changes, may not be applicable to patients with secondary or essential hypertension, given the differing pathophysiology and management approaches.

However, we acknowledge the importance of distinguishing between these conditions and will include a statement in the discussion highlighting that the recommended lifestyle interventions are tailored for prehypertension and should not be extended to secondary hypertension or essential hypertension without further evidence.

We have revised the introduction (Line 68-71) to clarify this point, and we hope this addresses your concern.

---

## [Editor Report · Decision Letter 3]

24 Sep 2024

Developing lifestyle intervention program for pre-hypertensive patients; consensus building using a modified Delphi approach

PONE-D-24-23230R3

Dear Dr. Hassan,

We’re pleased to inform you that your manuscript has been judged scientifically suitable for publication and will be formally accepted for publication once it meets all outstanding technical requirements.

Kind regards,

Yogesh Kumar Jain, MPH

Academic Editor

PLOS ONE
---

## [Editor Report · Acceptance letter]

2 Oct 2024

PONE-D-24-23230R3 

PLOS ONE

Dear Dr. Hassan, 

I'm pleased to inform you that your manuscript has been deemed suitable for publication in PLOS ONE. Congratulations! Your manuscript is now being handed over to our production team.

Kind regards, 

on behalf of

Dr. Yogesh Kumar Jain 

Academic Editor

PLOS ONE